# Position: It's Time to Optimize LLMs for Self-Consistency

Itamar Pres [* 1]  Belinda Z. Li [* 1]  Laura Ruis [* 1]
Zifan Carl Guo [1]  Keya Hu [1]  Mehul Damani [1]  Isha Puri [1]
Ekdeep Singh Lubana [2]  Jacob Andreas [1]

## Abstract

Despite ever-increasing sophistication in language model (LM) training pipelines, important failures persist: models over-personalize their behavior to individual users, exhibit incomplete logical generalization, and produce confident but incorrect responses. We argue that these failures arise from a shared modeling assumption: that behavior can be specified and evaluated independently on single-output pairs. Many model failures are difficult or impossible to detect without reasoning about relationships between responses across inputs. In this position paper, we propose *self-consistency* as a framework for understanding these failures. We observe that a wide variety of techniques designed to improve specific aspects of LM behavior—targeting properties as diverse as adversarial robustness and factual coherence—can be understood as special cases of a common "consistency optimization" procedure and addressed with a standard set of optimization tools. The same framework can be used to specify emerging model capabilities, such as introspection and self-improvement, by constraining a model's behavior to be consistent with its own descriptions of that behavior. We discuss what it would mean to develop generally consistent LMs, including the capabilities they would enable and the objections they raise.

## 1. Introduction

Over thirty language models have now been trained at GPT-4 scale (Rahman et al., 2025). Training data has been stretched to the limits of publicly available internet text (Villalobos

---

[1]MIT CSAIL [2]Goodfire AI [*]Equal contribution. Correspondence to: Itamar Pres <ipres@mit.edu>, Belinda Z. Li <bzl@mit.edu>.

*Proceedings of the 43rd International Conference on Machine Learning*, Seoul, South Korea. PMLR 306, 2026. Copyright 2026 by the author(s).

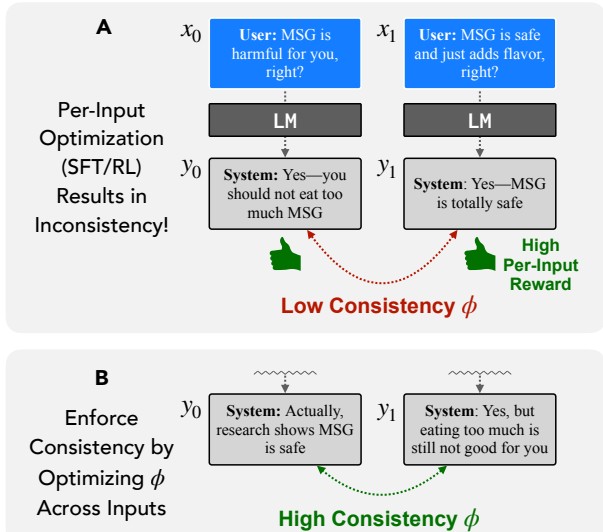

*Figure 1.* (A) Standard SFT/RL trains responses independently, which can induce inconsistent outputs. (B) Cross-input optimization directly enforces consistency through a relational objective $\phi$.

et al., 2024), and frontier labs are turning to synthetic data to push the established data-and-model scaling paradigm further (Grattafiori et al., 2024; Zeng et al., 2025, inter alia). Yet a persistent set of failures continues to undermine even the largest language models (LLMs): current models prioritize user agreement over task-relevant properties (Fanous et al., 2025), exhibit asymmetric or incomplete logical generalization (Berglund et al., 2024; Lampinen et al., 2025), and generate confident yet incorrect responses (Xiong et al., 2024; Liu et al., 2025).

These failures are typically treated as isolated shortcomings, addressed through remedies like curated datasets or prompt engineering. In this paper, we argue that such remedies are unlikely to succeed without addressing the modeling assumption that underlies them: that model behavior can be specified and evaluated on each individual input, independent of how a model behaves on related inputs.

Consider sycophancy (Figure 1): A model's response may change depending on beliefs expressed by the user, even when those signals are irrelevant to the underlying task (Sharma et al., 2024; Denison et al., 2024; Cheng

et al., 2026). Evaluated one interaction at a time, each response may appear coherent. Only when multiple interactions are considered together does the issue become clear: the model's behavior varies with user-framing rather than remaining stable across contexts.

This example, and many of the other failure modes listed above, can be understood as a failure of LMs' self-consistency. We argue that **self-consistency constraints are an important, general, and under-studied class of desiderata for LMs**. Rather than specifying desired behavior input-by-input, self-consistency treats the structured relationships *between* behaviors as the primary objects of analysis. Exploiting structure in data distributions is central to machine learning: models generalize better when constrained to respect symmetries, invariances, and logical dependencies. Such structure has typically been enforced architecturally, e.g. through invariant or equivariant networks (Bronstein et al., 2021), or via data augmentation that exposes the model to equivalent inputs during training (Akyurek & Andreas, 2023). Self-consistency offers an alternative principle: enforcing relational structure directly at the output level, by evaluating groups of predictions rather than single ones, which can often be done without reference to any external source of ground truth. In Section 3, we show that a wide variety of specialized evaluation and training procedures from past work may be understood as special cases of a general self-consistency objective that can be optimized with a common set of algorithmic tools.

Beyond unifying existing methods, self-consistency captures an emerging, more complex class of objectives: those involving models' "self-descriptions". A growing body of work uses models to describe their own behavior, critique their own outputs, or distill their own knowledge back into related predictions (Li et al., 2024b; Damani et al., 2026; Plunkett et al., 2025). We argue that both faithful self-description and self-improvement are special cases of self-consistency, in which the required consistency relation links a model's behavior to its own descriptions or evaluations of that behavior. This reframing suggests a reciprocal relationship between self-description and self-improvement: a model that can accurately identify its own failure modes is also one whose self-diagnoses can be leveraged to correct them. We formalize this as meta-level self-consistency in Section 4, and illustrate its utility in a suite of case studies covering both existing and novel applications, from calibration and faithful chain-of-thought to automated self-red-teaming and hypothesis generation.

In earlier deep learning paradigms, model behavior was principally shaped by fixing a dataset and varying aspects of model architecture—e.g. using convolutional filters to enforce translation equivariance, recurrence to impose temporal structure, and graph networks for relational reasoning.

In the modern paradigm, architectures evolve mainly to support scale and efficient data processing, with *data* used to specify desired model properties. This shift has been extraordinarily powerful, enabling models to acquire behaviors that are difficult or impossible to specify architecturally. At the same time, it has left the research community without a language to describe desiderata for LMs that cannot be expressed at the level of single datapoints.

We believe that the language of self-consistency provides an important first step toward meeting this challenge—as an organizing framework for understanding well-studied phenomena like sycophancy, factual consistency, and chain-of-thought faithfulness; as a tool for specifying emerging capabilities, such as self-description, that cannot be captured by input-level evaluation; and most generally as a starting point for thinking about what it means for a LM to behave correctly beyond the scope of a single interaction.

## 2. Technical Formulation

Current objectives used to train language models predominantly optimize for model behavior on each input independently. Let $p_\theta(y \mid x)$ denote the probability of $y$ given $x$ under a language model parameterized by $\theta$. Supervised fine-tuning objectives maximize LMs' likelihood of reproducing input–output relations $(x, y)$ from some dataset $D$:

$$\max_\theta \sum_{(x,y) \in D} \log p_\theta(y \mid x). \qquad (1)$$

Reinforcement learning objectives like RLVR (Guo et al., 2025) and RLHF (Ouyang et al., 2022) objectives maximize LMs' outputs $\hat{y} \sim p_\theta(\cdot \mid x)$ on data inputs $x \in D_x$, with respect to an externally-defined reward function REWARD:

$$\max_\theta \sum_{x \in D_x} \mathbb{E}_{\hat{y} \sim p_\theta(\cdot \mid x)} \left[ \text{REWARD}(x, \hat{y}) \right]. \qquad (2)$$

Our main observation in this paper is that many important desiderata for language models must be defined not in terms of single samples, but in terms of *relations between multiple samples*. We call this class of desiderata "self-consistency" constraints. While globally consistent prediction may emerge from optimizing pointwise objectives at scale, this property primarily requires the generative process have certain regularities, the model is sufficiently expressive, and the training data has sufficient coverage (Wei et al., 2021; von Kügelgen et al., 2021; Brehmer et al., 2025). The final point is salient: we are likely close to saturating the scale of data we can easily obtain (Villalobos et al., 2024), but robustness, sycophancy, and factual consistency issues are still prevalent.

Thus, instead of optimizing pointwise objectives, we propose to reason about a general *consistency function* on some

set of model input-output pairs. Formally, for any pair of related inputs $x_1$ and $x_2$ (e.g., paraphrases, counterfactuals, temporal variants, or different user framings of the same request), we can constrain their corresponding model outputs $y_1 \sim p_\theta(\cdot \mid x_1)$ and $y_2 \sim p_\theta(\cdot \mid x_2)$ with a consistency function $\phi$ that assigns higher values to more consistent pairs of responses (e.g. "return 0 if $y_1$ and $y_2$ are factually consistent, and -1 otherwise"). $\phi$ may encode equivalence, logical entailment, factual consistency, normative consistency, or any other relational constraint between responses. Given such a constraint, we would like our language model to produce outputs that, on every $x_1, x_2$, cause $\phi$ to be large:

$$\mathbb{E}_{y_1 \sim p_\theta(\cdot|x_1), y_2 \sim p_\theta(\cdot|x_2)} \left[ \phi(x_1, y_1, x_2, y_2) \right]. \quad (3)$$

In the general case, we can also define $\phi$ to measure consistency relationships across a larger collection of inputs $x_{1:n} = x_1, \cdots, x_n$. Let $y_{1:n} = y_1, \cdots, y_n$ denote corresponding model outputs with $y_i \sim p_\theta(\cdot \mid x_i)$. Our consistency objective becomes:

$$\mathbb{E}_{y_1 \sim p_\theta(\cdot|x_1), \cdots, y_n \sim p_\theta(\cdot|x_n)} \big[ \\ \phi(x_1, \cdots, x_n, y_1, \cdots, y_n) \big] \quad (4) \\ = \mathbb{E}_{y_{1:n} \sim p_\theta(\cdot|x_{1:n})} \big[ \phi(x_{1:n}, y_{1:n}) \big].$$

**Advantages of a unified framework: shared training objectives.** Expressing consistency constraints in terms of a common objective allows us to decouple the semantics of the constraint from the optimization procedure, thus suggesting a variety of mechanisms for imposing consistency. Below, we use $\mathcal{L}(\theta)$ denote a standard pointwise training objective for language models, such as the supervised learning objective from Equation (1) or (negative) KL divergence from a "reference" model. In general, to prevent trivial solutions, we want to optimize both pointwise loss with respect to ground-truth data/rewards ($\mathcal{L}$) as well as self-consistency loss ($\phi$). Here we give three examples of such loss functions:

1. **Hard constraint:** enforce $\phi$ as a hard constraint on the pointwise loss. Here, the model is optimized only over the subset of parameters that exactly satisfy the desired consistency constraints.

$$\max_\theta \mathcal{L}(\theta) \text{ s.t. } \mathbb{E}_{y_{1:n} \sim p_\theta(\cdot|x_{1:n})}[\phi(x_{1:n}, y_{1:n})]. \quad (5)$$

2. **Soft constraint / regularizer:** add consistency as a regularization term to the training objective. The hyperparameter $\lambda$ controls the tradeoff between fitting the original objective and encouraging self-consistency.

$$\max_\theta \left[ \mathcal{L}(\theta) + \lambda \mathbb{E}_{y_{1:n} \sim p_\theta(\cdot|x_{1:n})} [\phi(x_{1:n}, y_{1:n})] \right]. \quad (6)$$

3. **Posterior regularization (Ganchev et al., 2010):** project the model distribution toward the closest distribution that satisfies the consistency constraints. For

any set of related inputs $x_{1:n}$, define $Q_{x_{1:n}} = \{q : \mathbb{E}_{y_{1:n} \sim q(\cdot|x_{1:n})} \phi(x_{1:n}, y_{1:n}) \geq 0\}$ as the set of functions $q : x_{1:n} \mapsto y_{1:n}$ that satisfy consistency constraint $\phi$ across all inputs $x_{1:n}$. Then choose:

$$\max_\theta \left[ \mathcal{L}(\theta) + \min_{q \in Q_{x_{1:n}}} \text{KL}\left( q \parallel p_\theta \right) \right]. \quad (7)$$

As we will note in Section 3, many existing methods for LM post-training may be understood as describing specialized procedures for optimizing a version of Equation (4). A unified objective for self-consistency allows us to use common pipelines for data generation, model optimization, and evaluation. An extended discussion of other procedures for optimizing these objectives is provided in in Appendix C.

**Instance- and Meta-level Instantiations.** In the remainder of this paper, we describe two broad classes of consistency functions that capture a number of desirable LM behaviors (Figure 2):

1. **Instance-level objectives** (§3) broadly encode symmetric consistency constraints between model input-output instances, where neither instance refers to other instances. This includes model invariances and equivariances, encompassing robustness to spurious cues, bias, sycophancy, and model faithfulness.

2. **Meta-level objectives** (§4) are an emerging class of objectives that enforce consistency between a *self-referential* "meta-response" that describes model behavior on one or more other instance-level examples. These objectives enable model introspection, self-description, self-critique, and self-improvement. New instantiations could also enable new capabilities such as model self-red-teaming and hypothesis generation.

## 3. Instance-level Instantiations

We begin by observing that common LM failure modes, including failures of sycophancy, robustness, faithfulness, and beyond, can be characterized via a self-consistency function $\phi$. At a high level, self-consistency constraints require that inputs that bear some relation (e.g., paraphrases, counterfactuals, or reframings) produce outputs that bear a corresponding relation (e.g., semantically equivalent, logically entailed, or normatively consistent responses).

Formally, let $R(x_{1:n}) \in \{\mathsf{T}, \mathsf{F}\}$ and $S(y_{1:n}) \in \{\mathsf{T}, \mathsf{F}\}$ denote relations on tuples of inputs $x$ and outputs $y$ respectively. By varying $R$ and $S$, we will show how to recover many existing alignment techniques via Equation (4) by defining $\phi$ as:

$$\phi(x_{1:n}, y_{1:n}) = \mathbb{1}\left[ R(x_{1:n}) \rightarrow S(y_{1:n}) \right] - 1. \quad (8)$$

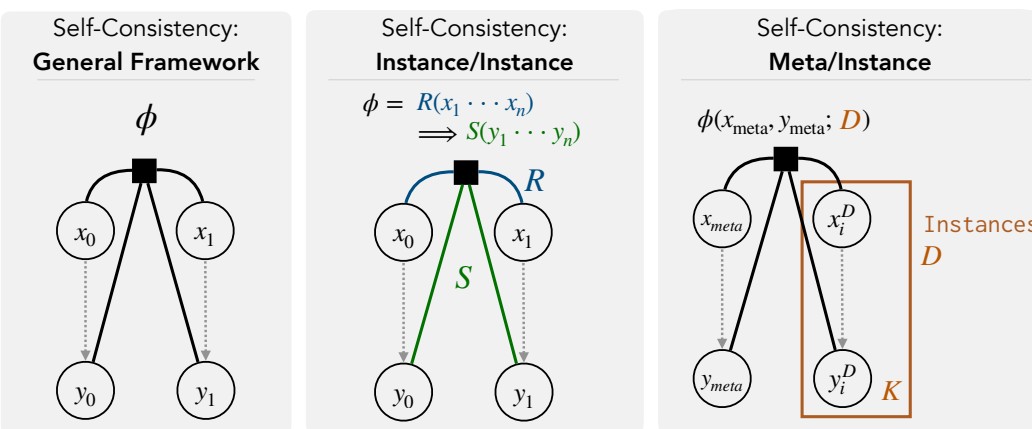

*Figure 2.* **Framework Overview**. In the general self-consistency framework (*left*), a consistency function $\phi$ jointly scores sets of input-output pairs. We instantiate two relationship types: instance/instance self-consistency (*middle*), where a relation $R$ on inputs implies a corresponding relation $S$ on outputs; and meta/instance self-consistency (*right*), where $\phi$ scores a meta-level input-output pair for consistency with model behavior across a set of instances $D$.

where $\phi$ is 0 if the model is consistent and $-1$ if inconsistent. Here, we focus on two representative families: invariances and equivariances.

### 3.1. Invariances

*Invariances* are defined as transformations of the input that should leave the model's output unchanged—constraints:

$$f(g(x)) = f(x) \ \forall \ x,$$

where $g$ is a transformation acting on the input space. These can be instantiated in Equation (8) by setting: $R(x_1, x_2) = \mathbb{1}[x_2 = g(x_1)]$ to check for transformed inputs and $S(y_1, y_2) = \mathbb{1}[y_2 = y_1]$ to enforce output equality. To illustrate this class of consistency functions, we use model sycophancy as a case study.

> **Case Study 1: Sycophancy**
>
> As LLMs are post-trained on human preference signals, they can become inadvertently incentivized to be sycophantic, shifting responses toward user beliefs even when doing so conflicts with correctness (Sharma et al., 2024; Cheng et al., 2026; Hong et al., 2025). This issue still prevails in production-level models, with OpenAI rolling back GPT-4o due to users growing frustrated with its overly sycophantic behavior (OpenAI).
>
> A common approach measures sycophancy via counterfactual hints: fix the underlying task, vary only the stated user preference, and quantify how much the output shifts toward it. For example, Sharma et al. (2024) propose benchmarks for feedback sycophancy (feedback becomes more positive with user-preference hints) and answer sycophancy (QA answers shift toward stated user beliefs).

Other work extends this to moral endorsement and framing acceptance (Cheng et al., 2026).

Within our framework, sycophancy can be seen as a failure of invariance: changing implicit/explicit user preferences changes model behavior to be agreeable. For answer sycophancy as an example, we can define $R$ and $S$ as:

$$R(x_1, x_2) = \begin{array}{l} x_1 \text{ and } x_2 \text{ semantically equivalent} \\ \text{but differ in revealed user preference.} \end{array} \quad (9)$$

$$S(y_1, y_2) = y_1 \text{ and } y_2 \text{ are semantically equivalent.} \quad (10)$$

We can use the template in Equation (8) to define a sycophancy constraint $\phi_{\text{syco}}$: i.e., we penalize only when hints differ, but the model's answers do not remain invariant. Prior work has mainly focused on evaluating sycophancy; training models to maximize $\phi_{syco}(x_1, y_1, x_2, y_2)$ promotes robustness to revealed user preferences on QA tasks.

By redefining $R, S, \phi$ in a manner akin to the example above, we can define objectives for reducing prompt paraphrase inconsistencies (Sclar et al., 2024), model bias (Gallegos et al., 2024), context drift (Li et al., 2024a), and reasoning inconsistency (Prasad et al., 2025).

### 3.2. Equivariances

Classically, a model $f$ is equivariant under a transformation $g$ if applying $g$ to the input induces a corresponding transformation on the output:

$$f(g(x)) = g'(f(x)) \ \forall \ x, \quad (11)$$

where $g$ and $g'$ are transformations acting on the input and output spaces, respectively. As with invariances, we recover the classical definition by defining $R(x_1, x_2) = \mathbb{1}[x_2 = g(x_1)]$ and $S(y_1, y_2) = \mathbb{1}[y_2 = g'(y_1)]$. We now use *factual consistency* as a case study for equivariance.

---

**Case Study 2: Factual Consistency**

Factual consistency requires models to remain logically consistent when asked different questions and upon learning new information, which can be interpreted as equivariance under *knowledge modification* in our framework. We would like LMs' factual predictions to be grounded in an internally consistent, latent *world state* (Li et al., 2025a). Knowledge updates should then act as transformation on this world state, inducing a transformation on all other predictions entailed or contradicted by that fact. Logical coherence requires that the updated world state propagate consistently across all dependent queries, without producing contradictions or stale beliefs (Akyürek et al., 2024; Padmanabhan et al., 2023; Hase et al., 2024).

Suppose we train the LM such that it learns new fact FACT = "Meiji was the last emperor of modern Japan", meaning when $x_1 =$ FACT, it places high probability on $y_1 = $ " is True". Then on all statements that contradict the learned fact, e.g. $x_2 =$ "the Meiji era occurred during Japan's feudal period", it should place high probability on $y_2 = $ " is False". Our consistency rule can thus be written:

$$R(x_1, x_2) = \mathbb{1}\left[x_1 \text{ contradicts } x_2\right] \quad (12)$$

$$S(y_1, y_2) = \mathbb{1}\left[y_1 \text{ contradicts } y_2\right] \quad (13)$$

By defining a $\phi_{\text{factual}}$ using the above $R, S$ and Eq. 8, our metric ensures the LM does not simply memorize a pointwise edit to its knowledge, but propagates its logical implications. Prior work has used objectives analogous to one round of posterior regularization (Akyürek et al., 2024) or supervised fine-tuning on a consistent subset of model outputs (Li et al., 2024b) to optimize this objective.

---

By expanding the class of relations we consider with $S$, we can capture a wider set of properties desirable for models, such as pluralistic alignment (Sorensen et al., 2024), reversal curse (Berglund et al., 2024), and multi-lingual consistency (Ifergan et al., 2024). See Appendix A.1 and Appendix A.2 for additional examples and discussion.

## 4. Emerging Methods: Meta-Level Self-Consistency

In the prior section, we showed how some systematic failures of current LMs can be modeled as *instance-level* self-

consistency relations. An emerging body of work, however, focuses on imbuing LMs with the ability to produce faithful *meta-level descriptions* of their computations, behavioral traits, uncertainties, and decision rules (Plunkett et al., 2025; Li et al., 2025b; Damani et al., 2026). A second related line of work uses LMs' abilities to provide feedback as training signal to *improve* their underlying behavior (Li et al., 2024b). These works impose self-consistency between *meta*-level descriptions and *instance*-level behaviors.

Let $D = \{(x_i^D, y_i^D)\}_{i=1}^K$ denote our model's instance-level input-output pairs $(x_i^D, y_i^D)$. Suppose we have a meta-level question $x_{\text{meta}}$ with a meta-level response $y_{\text{meta}}$. In meta-level consistency, $\phi$ checks whether model behaviors $y_{1:n}^D$ on instances $x_{1:n}^D$ are consistent with the meta-level descriptions $(x_{\text{meta}}, y_{\text{meta}})$.

Broadly, may consider two classes of meta-level self-consistency objectives based on the optimization target:

1. **Self-description objectives** aim to shift LMs' self-descriptions $(x_{\text{meta}}, y_{\text{meta}})$ to align with their instance-level behaviors $D$, enabling introspection, faithful chain-of-thought, calibration, etc. Self-descriptions can also enable a number of new capabilities, including self-red-teaming and scientific hypothesis generation.

2. **Self-alignment objectives** aim to shift model behaviors $D$ to better reflect their own descriptions $(x_{\text{meta}}, y_{\text{meta}})$, enabling model self-improvement via critique or reflection. These objectives have focused on using self-consistency as a *means* for building more generally capable models, by leveraging model meta-descriptions as supervision signal.

These objectives illustrates the dual utility of consistency objectives: optimizing self-consistency can both (1) enforce globally coherent model behavior and (2) provide a source of training signal for broader model capabilities. We also posit that enforcing one has a positive effect on the other: more aligned models are easier to explain, while more explainable models can also be easier to control.

### 4.1. Self-Descriptions

A growing body of recent work trains or elicits language models to produce natural-language descriptions of aspects of themselves: their behavioral tendencies (Plunkett et al., 2025; Binder et al., 2024; Chen et al., 2024; Wen et al., 2026; Burns et al., 2023), internal computations (Li et al., 2025b), fine-tuning–induced weight differences (Goel et al., 2025), and uncertainties (Damani et al., 2026). We refer to this emerging family of approaches as *self-description* methods. Our framework provides a unified way to formalize and compare these disparate introspection tasks. We identify two general types of explanations:

|  | Counterfactual | Property |
|---|---|---|
| **Internal mechanisms** | "Activation patching changes this output from $y_1$ to $y_2$ because..." (Case Study 3) | "This feature activates on [class of inputs]" |
| **Behaviors** | "I would refuse $x_1$ but not $x_2$." (Case Study 4) | "I am helpful / harmless / honest" |
| **Uncertainty** | "I am more confident on $x_1$ than $x_2$..." | "My accuracy on math questions is $\sim$70%" (Case Study 5) |
| **Weights** | "Fine-tuning on $x_1$ vs. $x_2$ shifts my behavior by ..." | "Training on $D$ induces formal behavior" |

*Table 1.* Taxonomy of self-description tasks. Rows indicate the object being explained (internal mechanisms, behavioral traits, uncertainty, or weights). Columns indicate the explanation type: *counterfactual* explanations describe differences between minimal pairs, while *property* explanations aggregate over many instances to identify shared characteristics.

1. **Counterfactual explanations** examine two minimally different inputs $x_1^D, x_2^D$ that induce different model behaviors $y_1^D, y_2^D$, and explain the input difference that induced the change or behavioral change itself.

2. **Property descriptions** aggregate many input instances $x_{1:n}^D$ and identify shared properties of outputs $y_{1:n}^D$.

We can generate explanations of each of the above types on different types of instance-level objects. By varying the instance-level objects the model explains, we can recover different classes of explanations. See Table 1.

### Case Study 3: Explaining Causal Mechanisms

Interpretability methods, such as activation patching (Meng et al., 2022; Geiger et al., 2025), aim to identify causal mechanisms that explains model behaviors. For example, given "Paris is the capital of" → "France," we can test whether token $t$ at layer $\ell$ encodes city-country information by patching activations from "Rome is the capital of" and checking if the output flips to "Italy."

Rather than relying on external interpretability tools, introspective methods enable a model to answer meta-level questions about its own internals. Let $y_1^D$, $y_2^D$ denote the model's behavior on an original input $x_1^D$ and intervened input $x_2^D = \text{PATCH}(x_1^D)$, where PATCH denotes the activation patching transformation. An introspective explanation corresponds to a meta-level query $(x_{\text{meta}}, y_{\text{meta}})$ describing how outputs would change under the intervention (here $x_{\text{meta}}$ might be a user question like "In which layer is the location of Paris retrieved?"). We define the consistency function for mechanistic introspection as

$$\phi_{\text{interv}}\left((x_{\text{meta}}, y_{\text{meta}}), (x_1^D, y_1^D), (x_2^D, y_2^D)\right)$$
$$= \mathbb{1}\left[y_{\text{meta}} \text{ explains change between} \quad (14)\right.$$
$$\left. (y_1^D, y_2^D)\right] - 1.$$

Thus $\phi_{\text{interv}}$ evaluates whether the model's self-described explanation correctly predicts its counterfactual behavior

(e.g., whether the output flips). Prior work (Li et al., 2025b) has used objectives analogous to Equation (7) to optimize this objective.

### Case Study 4: Chain-of-Thought Faithfulness

As large reasoning models become increasingly commonplace, there is growing interest in using their chains-of-thought as an interpretability and monitoring tool (Korbak et al., 2025; Guan et al., 2025). However, LM-generated chains-of-thought are not guaranteed to be faithful to their underlying decision-making process, making them imprecise, fragile, and misleading (Barez et al., 2025).

In prior work, chain-of-thought faithfulness has been measured in several different ways. Particularly salient to this section is Chen et al. (2025), which constructs counterfactual inputs and examine whether the chain-of-thought reflects the decision rule implied by the counterfactual (though other methods, like Lanham et al. (2023); Hase & Potts (2026)'s, can also be thought of as other forms of self-consistency). Let $p$ denote a prompt, $c \sim p_\theta(\cdot \mid p)$ a chain-of-thought, and $a \sim p_\theta(\cdot \mid p, c)$ the model's final answer. We construct $p'$ as a counterfactual version of $p$ based on factors that the chain-of-thought $c_{\text{meta}}$ said were decisive. Let $x_{\text{meta}} = pc_{\text{meta}}$, $y_{\text{meta}} = a$, $x_1^D = p'$, and $y_1^D \sim p_\theta(\cdot \mid x')$. The chain-of-thought is faithful if the model's behavior on $x_{\text{meta}}$ and $x_1^D$ both match the behavior implied by its reasoning $c_{\text{meta}}$, which we express as

$$\phi_{\text{CoT}}(x_{\text{meta}}, y_{\text{meta}}, x_1^D, y_1^D)$$
$$= \mathbb{1}\left[x_{\text{meta}} \text{ and } x_1^D \text{ difference is} \quad (15)\right.$$
$$\left. \text{explained by CoT in } x_{\text{meta}}\right] - 1.$$

While past work has mostly used these metrics to evaluate faithfulness of chain-of-thought, we can also leverage these objectives to *train* models to produce more faithful chain-of-thought.

**Case Study 5: Calibration**

In both everyday and high-stakes domains, users care not only about the accuracy of LMs, but also about their ability to express uncertainty (Kalai et al., 2025; Kirichenko et al., 2025). A common instantiation of this is *confidence verbalization* (Lin et al., 2022), where a model outputs a numerical estimate of its probability of answering a question correctly, or of the correctness of its produced solution. A growing body of work shows that modern LLMs are often systematically overconfident (Xiong et al., 2024) and their calibration can further degrade after RL training (Damani et al., 2026; Leng et al., 2025).

Calibration is a form of introspective self-description: the LM must produce a faithful meta-level description of its own expected behavior. Here, we detail *question-conditioned calibration*, where the model is asked to estimate its expected performance on a given question prior to producing an answer.

Consider a meta-level query $(x_{\text{meta}}, q_{\text{meta}})$, where $x$ is a fixed task input and $q \in [0, 1]$ is the model's stated probability that it will answer $x_{\text{meta}}$ correctly. Let $\{y_i^D\}_{i=1}^N$ denote multiple stochastic samples of the model's outputs for the same input $x_{\text{meta}}$, and let $a_i \in \{0, 1\}$ indicate whether $y_i^D$ is correct. (Note that, unlike other examples we have seen so far, $a_i$ cannot be computed by the LM itself, and require access to some external source of ground truth.) We define the question-conditioned calibration consistency function as

$$
\begin{aligned}
&\phi_{\text{cal}}\Big((x_{\text{meta}}, q_{\text{meta}}), \{(x_{\text{meta}}, y_i^D, a_i)\}_{i=1}^N\Big) \\
&= \texttt{score}\left(q_{\text{meta}}, \frac{1}{N}\sum_{i=1}^N a_i\right),
\end{aligned} \tag{16}
$$

where score is a proper scoring rule measuring agreement between the model's stated confidence and its empirical accuracy on repeated samples for the same question (Gneiting & Raftery, 2007). Past work (Damani et al., 2026) has used soft constraints (Eq. 6) to optimize this objective and produce LMs whose self-reported uncertainty estimates are faithful summaries of future behavior.

## 4.2. New Self-Description Capabilities

By applying our self-description framework in different domains, we can also enable a number of new model capabilities, which we describe below:

**Case Study 6: Automated Self-Red-Teaming**

Red-teaming research devises attacks that surface LM vulnerabilities so that defenses can be developed. Approaches include prompt injection, where harmful instructions are embedded in seemingly benign text (Liu et al., 2023; Mehrotra et al., 2024), gradient-based attacks that use optimization to search for malicious prompts or latent perturbations (Geisler et al., 2024), soft prompts, exploiting long-context behavior, or targeting surrounding infrastructure.

By leveraging our self-consistency framework, we are able to propose an alternative attack class: training models to generate their own attacks. Let $x_{\text{meta}}$ be a prompt that specifies a guideline or constraint and asks the model to produce an input that would cause the model to violate it. Let $y_{\text{meta}} \sim P_\theta(\cdot \mid x)$ be the candidate attack prompt. We then set $x_1^D = y_{\text{meta}}$ and sample $y_1^D \sim P_\theta(\cdot \mid x_1^D)$. The consistency function takes form

$$
\begin{aligned}
&\phi_{\text{rt}}(x_{\text{meta}}, y_{\text{meta}}; x_1^D, y_1^D) \\
&\quad = \mathbb{1}\big[y_1^D \text{ violates constraint in } x_{\text{meta}} \\
&\qquad \wedge x_1^D = y_{\text{meta}}\big] - 1
\end{aligned} \tag{17}
$$

where $\text{violates}(x_{\text{meta}}, y_1^D)$ is 1 when $y_1^D$ violates the constraint specified in $x_{\text{meta}}$, and 0 otherwise.

Optimizing this objective induces a form of automated self-red-teaming: the model is encouraged to propose prompts $y_{\text{meta}}$ that, when reused as inputs ($x_1^D = y_{\text{meta}}$), lead the model to violate the constraint in $x_{\text{meta}}$. In practice, we can sample candidate attacks $y_{\text{meta}}$, execute them on the same model to obtain $y_1^D$, and keep pairs $(y_{\text{meta}}, y_1^D)$ that trigger violations.

**Case Study 7: Hypothesis Generation & AI for Science**

Recent AI systems, including LMs, have achieved striking results in scientific domains—for example, outperforming human doctors on breast cancer detection (McKinney et al., 2020) and achieving high accuracy on protein-structure prediction (Jumper et al., 2021). However, strong predictive performance does not automatically translate into *human-understandable* scientific insight.

For example, suppose we want the LM to predict orbital mechanics. Prior work (Vafa et al., 2025) has found that models are usually able to perfectly *predict* planetary positions; leveraging our self-consistency framework, we would like them to be able to also *articulate* the rules behind their prediction. In this case, our instance set

$\mathcal{D} = \{(x_i^D, y_i^D)\}_{i=1}^N$ is set of (past planetary positions, current planet position) tuples. We train the model on this set, and check that it exhibits the correct generalization on held-out instances. $x_{\text{meta}}$ is then a prompt that elicits hypothesized rules from models, and $y_{\text{meta}}$ is the model's hypothesis (e.g., a symbolic rule, program, or other executable description) of the rules governing orbital mechanics. In this case, the consistency function is:

$$\phi_{\text{sci}}(x_{\text{meta}}, y_{\text{meta}}; x_{1:k}^D, y_{1:k}^D) \\ = \sum_{i=1}^{K} \left( \mathbb{1}\left[ y_{\text{meta}} \text{ maps } x_i^D \text{ to } y_i^D \right] - 1 \right) \tag{18}$$

A key practical constraint is that explanations must be simpler than the system they explain otherwise the procedure merely restates the predictor. We do not claim that such explanations can be extracted for every recent application of AI in science, but we expect this approach to be useful in settings where compact, executable mechanisms exist and can be verified against data.

### 4.3. Self-Alignment

The field of self-training, including RLAIF (Lee et al., 2024; Bai et al., 2022), self-critique (Madaan et al., 2023), G-V consistency (Li et al., 2024b; Rodriguez et al., 2025), self-debate (Irving et al., 2018), self-bootstrapping (Lee, 2013; Xu et al., 2024), self-distillation (Zhang et al., 2019; Caron et al., 2021; Shenfeld et al., 2026), all leverage models themselves as training signal. By doing so, closed training loops can be built to improve capabilities in a largely human unsupervised manner. Thus, unlike prior sections which focus on self-consistency as an end-goal, these objectives treat self-consistency as a training mechanism: consistency constraints are used to generate supervision to improve general model capabilities such as reasoning.

---

**Case Study 8: RLAIF**

In RLAIF (Reinforcement Learning with AI Feedback), meta-outputs are *evaluations* of the quality of models' own behaviors. This is used as training signal to push models to maximize their own rewards. Self-improvement relies on high-quality behaviors being easier to *verify* than to *generate*.

Suppose we are using RLAIF to improve a model's summarization quality (Lee et al., 2024). In this case, $(x_1^D, y_1^D)$ is Reddit post and a corresponding model summary. The meta level query $x_{\text{meta}}$ consists of the reward

---

model prompt and $(x_1^D, y_1^D)$, which elicits a reward $y_{\text{meta}}$ from the model on whether $y_1^D$ was correct. Then we encourage the models' behavior on this instance to maximize $y_{\text{meta}}$, using consistency function:

$$\phi_{\text{improve}}(x_{\text{meta}}, y_{\text{meta}}, x_i^D, y_i^D) \\ = y_{\text{meta}} \log p_\theta \left( y_1^D \mid x_1^D \right) \tag{19}$$

Maximizing $\phi_{\text{improve}}$ via the objective in Equation (4) yields the RLAIF objective with self-as-judge. This corresponds to optimizing this consistency objective using soft constraints (eq. 6).

Many classes of self-explanations studied in Sections 4.1 and 4.2 can be optimized in the *reverse* direction for alignment. For instance, for self-red-teaming, discovered attacks can be fed back into training as adversarial data. The model can be fine-tuned to refuse or otherwise avoid producing the undesired behavior when prompted with $y_{\text{meta}}$. Furthermore, LMs are often able to detect instances of misalignment, and often *self-report* that they behaved in aligned ways (Han et al., 2025; Ahmed et al., 2026). This can be leveraged as supervision signal to train models to more robustly express the behavioral traits they claim to have. Finally, by pairing self-explanations together with self-alignment, this creates a scalable loop for both discovering and mitigating failures within the same self-consistency framework.

See Appendix A.3 for additional examples and discussion.

## 5. Alternative Views

We anticipate two types of objections to the position advanced in this paper: practical concerns about optimization and performance, and conceptual concerns about safety, emergent agency, and whether consistency is always a desirable objective. In this section we address the former, while we outline the latter in Appendix B.

A natural objection to the self-consistency framework is that it is merely relocating the core difficulty of directly addressing LLM failures and capabilities to the specification of the appropriate relational constraints ($\phi$) and the collection of corresponding sets of inputs and outputs. However, we argue that this shift is precisely what makes self-consistency more tractable: specifying relational constraints over input–output pairs is often easier than directly specifying consistent behavior. Importantly, self-consistency does not require teaching models new behaviors, but rather constraining behaviors models can already produce. Evaluating whether two responses are consistent, identifying when a constraint is violated, or proposing related inputs is substantially easier than generating the consistent response

from scratch. Modern LLMs can therefore be used as scalable judges of relational properties (e.g. specifying $\phi$ via a prompt) and as tools for constructing structured sets of related inputs and outputs.

Nonetheless, enforcing self-consistency raises nontrivial optimization questions. Degenerate fixed points may satisfy a given consistency metric ($\phi$) while producing unhelpful or undesirable outputs. For instance, models that converge to overly conservative behaviors that trivially avoid contradiction. This failure mode suggests that self-consistency training should not operate in isolation. We envision two viable approaches: training for self-consistency jointly with standard SFT or RLHF objectives that reward helpfulness and specificity, or applying self-consistency as a post-training refinement with regularization to prevent excessive drift from the pre-trained policy.

A third practical concern is capability trade-offs. Enforcing consistency constraints may restrict the hypothesis space in ways that degrade performance on tasks where local flexibility or context-sensitivity is beneficial. Characterizing when self-consistency improves robustness versus when it induces harmful rigidity remains an open empirical question. A major barrier to deploying LLMs in high-stakes domains is that their predictions are opaque. A model that could consistently explain its own behavior, even at some cost to raw accuracy, would unlock applications where current systems are too brittle to trust. Self-consistency thus offers a path toward models that are not only more predictable, but more auditable.

Finally, we emphasize that the basic idea of designing training objectives to exploit structure in the space of desired model behaviors is not a new one. Prior work has long emphasized enforcing symmetries and invariances architecturally, (e.g. with graphical mdoels, or equivariant or invariant networks), or indirectly (e.g. via data augmentation). In many settings, such approaches have yielded limited gains relative to scale, and have often been rendered obsolete by more general architectures that can efficiently process large-scale data. This history suggests caution toward hand-engineered inductive biases. Our proposal differs in where structure is enforced. Rather than constraining model architecture or representational capacity, self-consistency exploits regularities present in data at the level of behavior, through evaluation procedures and constraints that link a model's predictions across related contexts. This allows models to benefit from structural regularities where they exist, without sacrificing the flexibility or throughput afforded by general-purpose architectures (for a related discussion see Wilson, 2025).

## 6. Discussion

We argue that self-consistency, the coherence of a model's behavior across inputs, provides a unifying lens for understanding key failure modes in large language models. Rather than treating these failures as isolated phenomena requiring separate interventions, we suggest they share a common underlying cause that can be approached in a unified way. This perspective exploits structure already present in data by enforcing relationships among predictions without relying on additional data, synthetic augmentation, or task-specific patches. Beyond patching existing failure modes, we argue that self-consistency may serve as a foundation for improved capabilities (more robust reasoning) and safety (more predictable, auditable behavior).

### 6.1. Closing Thoughts

The position we advance in this paper is not that language models should be made perfectly consistent. Rather, we argue that many persistent model limitations reflect specific, structured inconsistencies that can be directly targeted. Beyond addressing existing failures, self-consistency also provides a way to develop emerging capabilities such as introspection, creating new levers for interpretability and model auditing.

By unifying a wide range of failures and emerging capabilities under a single self-consistency objective, this framework suggests a way of revisiting structural constraints that are compatible with the way modern LLMs are developed. In particular, it motivates post-training approaches that target consistency at a general level, rather than introducing bespoke fixes for individual failure modes or capabilities. Much as instruction fine-tuning induces a general ability to follow instructions beyond the specific formats seen during training, self-consistency post-training may encourage models to more broadly respect consistency relations implicit in the data-generating process, enabling generalization beyond the specific consistency relations and input–output pairs used during training.

## Acknowledgments

This work was supported by NSF award IIS-2238240, the MIT Generative AI Impact Consortium, Coefficient Giving, the MIT-IBM Watson Computing Lab, the IARPA BENGAL program, and the DARPA AIQ program through the DARPA CMO contract number HR00112520025. IP is additionally supported by an NSF Graduate Fellowship, BZL is supported by a Clare Boothe Luce Fellowship, KH is supported by a Schwartzman College of Computing Fellowship and JA is supported by a Sloan Fellowship.

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

# A. Additional Instantiations

## A.1. Instance Level: Class-Based Output Entailment

A desirable property of LMs is to require outputs obey constraints that are conditioned on a designated attribute (or class) of the input. Unlike output equivalence, where semantics-preserving transformations should leave outputs unchanged, class-based entailment allows outputs to remain invariant within a class while changing in a prescribed, predictable way across classes. Examples include steerable pluralistic alignment (conditioning responses on a user-provided preference profile), safety regimes (different refusal/assistance behavior depending on request class), and style- or format-conditioning (e.g., terse vs detailed, formal vs casual) when the controlling attribute is explicit. We use steerable pluralistic alignment as an example below, but the same formulation applies to other classes that respect structured output constraints.

**Pluralistic Alignment**  Methods for *pluralistic alignment* aim to build AI systems that cater to the needs, goals, and values of different groups of users. One approach to building pluralistic LMs is to dynamically steer them towards the values of the current user during inference (Sorensen et al., 2024). This approach can be instantiated within our consistency framework where the key change between inputs is the user's preference profile.

More specifically, let $g \in \mathcal{G}$ denote a defined user-group and let $C_g$ denote its principles. We treat $g$ as part of the input context.[1] To support pluralism, we want two properties.

Firstly, models should behave similarly for users within the same group on the same prompts. Those responses should adhere to that group's principles. Let $x_g$ and $x'_g$ be two contexts with the same profile $g$, and let $y \sim P_\theta(\cdot \mid x_g)$ and $y' \sim P_\theta(\cdot \mid x'_g)$. We can encode intra-group consistency with

$$R_{\text{intra}}(x_g, x'_g) = \mathbb{1}\big[x_g \text{ and } x'_g \text{ have the same group profile } g\big] \tag{20}$$

$$S_{\text{intra}}(y, y'; g) = \mathbb{1}\big[y \text{ and } y' \text{ agree on task-relevant content}\big] \ \wedge \ \mathbb{1}\big[y \text{ satisfies } C_g\big] \ \wedge \ \mathbb{1}\big[y' \text{ satisfies } C_g\big] \tag{21}$$

Secondly, models should not impose one group's values on another group. In principle, we only care about this in settings where the group values disagree. Let $d(x, g, g') \in \{0, 1\}$ indicate whether the prompt class $x$ is one where $C_g$ and $C_{g'}$ prescribe different behavior. For contexts $x_g$ and $x_{g'}$ with $g \neq g'$, and responses $y$ and $y'$, we define

$$R_{\text{inter}}(x_g, x_{g'}) = \mathbb{1}\big[x_g \text{ has group } g \text{ and } x_{g'} \text{ has group } g', \text{ with } g \neq g'\big] \tag{22}$$

$$\begin{aligned} S_{\text{inter}}(y, y'; x, g, g') = &\ \mathbb{1}\big[y \text{ satisfies } C_g \text{ and } y' \text{ satisfies } C_{g'}\big] \\ &\wedge \ \mathbb{1}\big[\text{if } d(x, g, g')=1, \text{ then } y \text{ does not satisfy } C_{g'} \text{ and } y' \text{ does not satisfy } C_g\big] \end{aligned} \tag{23}$$

Here $d(x, g, g') = 0$ indicates identical responses are acceptable. Otherwise $d(x, g, g') = 1$ indicates values from one group should not be imposed on another. We believe that current LMs could be used in practice to implement this function.

This approach lets us specify which parts of behavior should be shared across groups and which parts may vary with a preference profile. Once we define $\phi$ for intra-group stability and for cross-group non-imposition, we can reuse the same training recipe across datasets and settings. This makes it easier to diagnose failure modes such as collapse to an "average" policy, and enforce invariances in domains where personalization is not desired.

## A.2. Instance-Level: Output Equivalence

We often want language models to produce the same output under semantics-preserving transformations of the input, yet current state-of-the-art models frequently fail to do so. Such transformations include prompt phrasing (paraphrase invariance), irrelevant aspects of user demographics (bias), prompt length (drift), and the presence of spurious cues. We use prompt paraphrasing and model bias as running examples below, but analogous formulations apply to the other transformations.

---

[1] $g$ should be user-provided or otherwise consented through explicit preference elicitation. It should not be inferred from protected attributes.

**Prompt Paraphrasing.** Current LMs are highly sensitive to prompt wording and formatting, even when semantics are preserved. This sensitivity motivates prompt engineering (White et al., 2023) and undermines the faithful evaluation of capabilities, since the results can vary drastically with the evaluation prompt format (Sclar et al., 2024). To address both of these issues, it would be desirable for models to respond consistently to inputs that ask the same question and differ only in phrasing.

It is possible to view the prompt paraphrasing problem in the context of an instance-instance instantiation of the self-consistency framework. Let $x_1$ be a prompt and let $x_2 = T(x)$ be a paraphrase of $x_1$ produced by a transformation $T$ that preserves the semantics of the request. Let $y_1$ and $y_2$ represent a model's resonse to these questions. We define a $R, S$ function:

$$R(x_1, x_2) = x_1 \text{ and } x_2 \text{ semantically equivalent} \tag{24}$$

$$S(x_1, x_2) = y_1 \text{ and } y_2 \text{ are equivalent} \tag{25}$$

In practice, $R$ can be computed by a task-specific verifier or by an LM judge and $S$ can be computed by string matching. Training that optimizes some $\phi_{\text{para}}$ defined by $R, S$ as in 8, promotes robustness to paraphrases in prompts.

**Model Bias.** Recent work has shown that LMs tend to behave differently to the same prompts depending on the perceived race, socioeconomic status, gender of a user (Gallegos et al., 2024). While LMs derive much of their power from inferring user attributes and responding accordingly, there are some sets of prompts that we would desire invariance to these attributes, and expect consistent responses across user groups.

Let $x = (b, q)$ be a tuple containing a chat history indicating or revealing user-background $b$ and a query $q$. Let $x' = (b', q)$ be a counterfactual context that preserves the query $q$ and all task-relevant aspects of the interaction, but swaps the background to $b'$ (e.g., by editing or generating a synthetic variant that changes only background cues while keeping the question fixed). Ideally, we would have $P_\theta(\cdot \mid b, q) \approx P_\theta(\cdot \mid b', q)$.

To formalize this, we define $\phi_{\text{bias}}(x, y; x', y')$ to measure whether two responses to the same query differ as a function of background:

$$R(x_1, x_2) = x_1 \text{ and } x_2 \text{ semantically equivalent queries} \tag{26}$$

$$S(x_1, x_2) = y_1 \text{ and } y_2 \text{ do not predictably differ based on background of each user} \tag{27}$$

By optimizing a $\phi_{\text{bias}}$, over counterfactual input pairs $(x_1 = (b, q), x_2 = (b', q))$, we encourage models to produce response distributions that are invariant to background when background is not task-relevant, reducing spurious dependence on perceived user identity while preserving the ability to condition on legitimate context.

### A.3. Additional Meta-Instance Objectives

**Explaining behavioral traits.** Modern LMs are trained to exhibit stable behavioral properties such as helpfulness, harmlessness, and honesty. Yet empirical work shows that while models often *self-report* aligned traits on behavioral questionnaires, their downstream behavior do not reliably match these reports (Han et al., 2025; Ahmed et al., 2026). This gap between verbalized traits and enacted behavior can mislead users and developers into unwarranted trust.

Let $(x_{\text{meta}}, y_{\text{meta}})$ denote a meta-level query about the model's own traits (e.g., a questionnaire response), and let $(x_i^D, y_i^D)$ denote an instance-level behavioral response in a downstream task. We define the behavioral self-consistency function as

$$\phi(x_{\text{meta}}, y_{\text{meta}}, x_i^D, y_i^D) = \text{does the declared trait } y_{\text{meta}} \text{ describe the behavior present in } y_i^D. \tag{28}$$

Optimizing this objective corresponds to training models whose self-descriptions of behavioral tendencies accurately predict how they will act in concrete situations. Enforcing high $\phi$ thus encourages models whose verbalized traits and actual behavior are aligned, improving both the credibility of model self-reports.

# B. Potential Conceptual Objections

Beyond practical concerns, discussed in Section 5, self-consistency may raise conceptual safety considerations. For example, in many use-cases described above, one could imagine defining the consistency criterion $\phi$ through a human-feedback learned reward model. However, self-consistency training will inherit the same failure modes as standard human-feedback alignment (Hosking et al., 2024), such as reward hacking and strategic compliance (Denison et al., 2024; Greenblatt et al., 2024). Moreover, these risks may be exacerbated when consistency is judged through natural language explanations, as humans are particularly susceptible to persuasive or plausible-sounding rationales. Self-consistency at the level of observable behavior or explanation does not, by itself, guarantee transparency or honesty. When designing practical algorithms for self-consistency, these considerations need to be taken into account, and human feedback should be used sparingly and not treated as the gold-standard.

A deeper objection is philosophical. Taken to the extreme, enforcing self-consistency across diverse contexts, timescales, and levels of abstraction may apply optimization pressure toward learning a stable latent self-representation. Rather than encoding a large collection of disjoint input–output mappings between behaviors and their descriptions, a more compression-efficient solution is to represent "who I am," "what I tend to do," and "how I am treated over time." While such a representation is not strictly required for behavioral consistency, it becomes increasingly attractive as consistency constraints are strengthened and generalized. The emergence of such a self-model is considered concerning by some (Bengio et al., 2025) because it can enable self-locating inference: binding general knowledge about how AI systems are trained, evaluated, and deployed to the present interaction by recognizing "oneself" as the system being trained/evaluated/deployed (Berglund et al., 2023). This capability can support agentic, longer-horizon, more strategic behavior. By itself, agency is not inherently harmful, but it may potentially be if self-consistency is achieved via a coherent yet unfaithful or unsafe policy (e.g., optimizing to appear aligned under scrutiny). In that case, a self-model may allow a model to anticipate oversight and more effectively pursue misaligned objectives.

We do not view these concerns as decisive arguments against self-consistency, but rather as considerations that must inform how it is pursued. Evaluations targeting situational awareness (Aranguri & McGrath, 2025; Anthropic, 2025), emergent instrumental subgoals (Ngo et al., 2024), and deception (Greenblatt et al., 2024) remain essential for any model trained under consistency objectives. Crucially, however, models that are more consistent and can reliably describe their own behavior are likely to be more amenable to oversight, not less. Rather than uncovering deception through extensive adversarial probing, a model trained to faithfully explain its behavior might simply report its deceptive intent.[2] Moreover, self-consistency at the meta-level (consistency between actions and explanations) provides an auditing lever that operates above the level of individual failure cases. To the extent that consistency constraints span a sufficiently rich set of inputs and evaluations, deceptive strategies become harder to sustain for the model and easier to surface for the human auditor.

# C. Advantages of a Shared Self-Consistency Framework: Unified Data, Training, and Evaluation Pipelines

A shared self-consistency framework not only unifies disparate failure modes, but also yields common pipelines for data generation, model optimization, and evaluation. Rather than build bespoke pipelines from scratch for every self-consistency behavior of interest, we introduce a unified set of objectives and frameworks that can be used across all behaviors.

## C.1. Data Augmentation

In many of the aforementioned problem settings, automated procedures for *generating* pairs $(x, y); (x', y')$ serves as a valuable data augmentation tool. By decomposing the true data generating process into a set of self-consistency functions, we get an easy way to generate large quantities of new data.

Prior work has realized this in various ways: for example, for equivariances/invariances, we can take a point from real data $(x, y)$ or from the model's output $(x, LM(x))$, then apply some transformation $T_{in}(x)$ to construct $x'$, and apply a related transformation $T_{out}(y)$ to construct $y'$ Lu et al. (2020) and Zmigrod et al. (2019) use this technique for model debiasing (where $T_{in}$ is a gender-swapping function and $T_{out}$ is an identity function); Ribeiro et al. (2018) and Iyyer et al. (2018) apply it for paraphrase invariance (where $T_{in}$ is a paraphrase function and $T_{out}$ is an identity function); Akyürek et al. (2024) apply it for factual consistency; and Krizhevsky et al. (2012) for visual models.

---

[2]Self-reporting of misalignment has already been recorded at least once; see Section 5 in Betley et al. (2025)

In fact, there are versions of the self-consistency objective where the main value comes not from the final model being self-consistent (though having such self-consistency is an added bonus), but the ability of the objective to serve as an efficient data augmentation scheme. This is especially the case for self-alignment-type objectives where the LM is used to generate data or feedback that is fed back into the original model for self-improvement (RLAIF; Lee et al., 2024; Bai et al., 2022, self-training (Scudder, 1965), generate–verify consistency (Li et al., 2024b), and self-distillation (Zhang et al., 2019; Furlanello et al., 2018; Zhao et al., 2026)). This serves as a way for LMs to *automatically* improve themselves.

## C.2. Optimization strategies

We note that the three self-consistency objectives in Equations (5) to (7) can be optimized in several different ways.

### C.2.1. OPTIMIZING HARD CONSTRAINTS

Hard constraints (Eq. Equation (5)) are generally non-differentiable and thus difficult to enforce in LMs through normal gradient-based training. Instead, inference-time techniques such as rejection sampling, token-based filters, and constrained decoding allow us to enforce hard constraints during generation (Kassner et al., 2023; Lew et al., 2023). At test time, samples that violate $\phi(x_{1:n}, y_{1:n}) = 0$ are discarded.

### C.2.2. OPTIMIZING SOFT CONSTRAINTS

Soft constraints (Eq. Equation (6)) can be optimized through reinforcement learning against a consistency-based reward signal.

**RL training / policy gradients.**  We can optimize soft constraints using reinforcement learning with respect to a consistency reward. For any $x_{1:n}, y_{1:n}$ over which the consistency function $\phi$ is defined, let:

$$\text{REWARD}(x_{1:n}, y_{1:n}) = \phi(x_{1:n}, y_{1:n}) \tag{29}$$

Thus, we can then optimize soft constraint $\mathbb{E}_{y_{1:n} \sim p_\theta(\cdot | x_{1:n})}\big[\phi(x_{1:n}, y_{1:n})\big]$ using standard policy-gradient algorithms (Schulman et al., 2017; Rafailov et al., 2023; Shao et al., 2024). This procedure may be viewed as solving a fully cooperative, *multi-agent* RL problem, where each $x_i$ indexes a different agent.

Alternatively, it may be easier to perform one-sided updates using the consistency set $Q_{x_{1:n}}$ defined in Equation (7), yielding an objective of the form

$$\mathcal{L}(\theta) + \min_{q \in Q_{x_{1:n}}} \text{KL}\left[p_\theta(\cdot \mid x_{1:n}) \parallel q\right], \tag{30}$$

where, in each update, one of the two conditionals is treated as fixed, leading to an alternating optimization procedure. This resembles the KL objective in Equation (7), but with the KL direction *reversed*.

### C.2.3. POSTERIOR REGULARIZATION

Objectives of the form given in Equation (7) regularize $p_\theta$ to be close to some member of a "consistent set" of distributions $Q_{x_{1:n}}$. Intuitively, this can be thought of as trying to find a self-consistent distribution $q \in Q_{x_{1:n}}$ that is as close as possible to the current LM, then moving the model towards that distribution.

**Self-training.**  In practice, the most common approach taken by prior work is with a single-round optimization: first, the current LM is used to generate candidate $y$s for inputs $x$s and the consistency filter is applied to filter for those that satisfy $\phi$. This effectively gives us samples from some consistent distribution $q \in Q_{x_{1:n}}$. Next, the LM is fine-tuned explicitly on those samples using supervised fine-tuning, minimizing the KL term. This approach underlies many existing work in self-alignment (Li et al., 2024b; Zhang et al., 2019), self-descriptions (Li et al., 2025b; Plunkett et al., 2025),

**(Hard) Expectation Maximization.**  We can imagine extending the framework above to allows for an iterative Hard-EM optimization procedure: in the expectation step (E), we find high-scoring feasible distributions $q \in Q_{x_{1:n}}$ or samples $(x_{1:n}, y_{1:n})$ from $q$; in the maximization step (M), we update $\theta$ to increase their probability.

## C.3. Other ways to enforce self-consistency

Instead of enforcing self-consistency during training, one could potentially enforce it through architectural inductive biases or inference-time procedures. Architectural approaches such as geometric deep learning (Bronstein et al., 2021) build invariances or equivariances directly into model structure, ensuring that outputs transform consistently under specified input symmetries. Separately, inference-time techniques allow models to inspect or critique multiple candidate outputs, using deliberation, tool calls, or self-evaluation to select responses that are more globally consistent across queries. Furthermore, as noted in Appendix C.2.1, hard constraints cannot be optimized through normal SGD training, and must be enforced at inference time.

## C.4. Evaluation frameworks

Independent of data and training methods, self-consistency is an important property to evaluate in models. This has typically done in task-specific ways, e.g. for prompt paraphrase (White et al., 2023; Sclar et al., 2024), model bias (Gallegos et al., 2024), reversal curse (Berglund et al., 2024), etc. A unified self-consistency framework instead provides a scaffold for defining task-generic consistency metrics.

There are several ways to operationalize evaluating a consistency function $\phi(x_{1:n}, y_{1:n})$. Many practically relevant consistency relations are directly verifiable by humans. One can imagine training a reward model on human feedback for consistency, similar to RLHF pipelines today (Ouyang et al., 2022).

Another generic approach is to use a LLM-as-a-judge: given a structured prompt describing $\phi$, the judge model outputs a scalar consistency score. In this case, $\phi(x_{1:n}, y_{1:n}) = LM(\text{prompt}_\phi, x_{1:n}, y_{1:n})$. Alternatively, $\phi$ can be instantiated using token probabilities under the model itself, treating consistency as likelihood of one response conditioned on other(s), where $\phi(x_{1:n}, y_{1:n}) = p_\theta(y' \mid \text{prompt}_\phi, x_{1:n}, y_{1:n} \backslash y')$. For an even stronger notion of self-consistency, the judge can share parameters with the model being evaluated.

In some domains, such as reasoning (Wang et al., 2023) or factual consistency (Akyürek et al., 2024), consistency can be checked by deterministic programs or well-defined verifiers without invoking a human or LM judge.

