# OpenReview forum: "Position: It’s Time to Optimize LLMs for Self-Consistency"
_ICML.cc/2026/Position_Paper_Track — ICML 2026 Position Paper Track regular_

### Official Review · Reviewer_MvLJ · 2026-03-11

**Significance:** 3
**Argument Clarity:** 2
**Rating:** 5
**Confidence:** 3

**Questions:**

1. The paper attributes various seemingly different LLM errors (such as pandering, logical fallacies, and false confidence) to self-consistency violations.
Is this merely a retrospective, generalized explanation that "encompasses" many phenomena without truly revealing their common causes?
Does the framework have explanatory power, or is it just a conceptual convenience?
Does it distinguish between different types of errors, or does it lump all problems under the umbrella of "self-consistency"?

2. Does self-consistency require absolute consistency, or should the model be allowed to make reasonable adjustments in different contexts?
The problem is that some inconsistencies are indeed harmful inconsistencies, while others are reasonable and necessary (contextual adaptations). Which attributes should be included in the definition of self-consistency? Which should not be considered violations of self-consistency?

3. The paper criticizes modern LLM training as ``point-by-point optimization''.
However, LLM training is no longer entirely point-by-point optimization; for example, there are adversarial training, RLHF (reinforcement learning from human feedback), and multi-turn contextual optimization.
Therefore, the paper's criticism may oversimplify the actual training paradigm.

**Alternative Views Section:**

Yes

**Compliance With Llm Reviewing Policy A Conservative:**

Affirmed.

**Discussion Potential:**

3

**Final Justification:**

Thanks for the rebuttal. The authors have adequately addressed my concerns, and I have no further questions at this stage. I am inclined to accept.

May be can be seen as a minor suggestion, it would be beneficial to include a more detailed and in-depth discussion of the methods that are not entirely consistent with the so-called "point-by-point optimization", as the authors have mentioned in the response.

**Paper Summary:**

This paper points out that current LLMs suffer from several systemic problems, including sycophancy, insufficient logical generalization, and confident but incorrect answers.
The authors argue that these problems stem from the existing training paradigm's assumption that models can be trained and evaluated independently using single input-output pairs, neglecting the consistency of answers across different inputs.
The paper's main contribution lies in proposing **self-consistency** as a unified framework, formalizing and classifying it as follows:
Instance-level self-consistency: Focuses on whether the model's behavior remains unchanged or satisfies equivariance under different cues.
Meta-level self-consistency: Focuses on whether the model's explanations, confidence levels, or self-descriptions are consistent with actual behavior.
The authors illustrate the application of these concepts with concrete examples (such as sycophancy and factual consistency) and explore potential future capabilities, including self-reflection, credible explanations, self-red team testing, and hypothesis generation.
The paper's strength lies in providing a unified theoretical framework that allows various existing methods to be understood as special cases of consistency optimization, and it proposes a future research direction: making large language models more predictable, interpretable, and auditable by optimizing self-consistency.
The paper's limitation is that it does not provide new experimental algorithms or validation results; the proposed applications and capabilities are mostly potential hypotheses. Overall, the paper's value lies primarily in guiding the theoretical framework and research direction, rather than in actual performance improvements.

**Position:**

Yes

**Position In Title:**

No

**Related Work:**

3

**Strengths And Weaknesses:**

**Strengths**:

1. The stance is clear.

2. This paper tries to incorporate the challenges faced by various LLMs into a unified framework, and on this basis, provides technical formalization and several optimization strategies.


**Weakness**:

1. From the title, I cannot identify that the paper tries to point out the issues for LLMs.

2.  The discussion in the paper on “when consistency should not be pursued” lacks sufficient detail.

3. The author emphasizes that a unified framework can optimize multiple model properties simultaneously, but these goals may conflict in practice.

**Support:**

3

---

> ### Author Rebuttal · Authors · 2026-03-31
>
> We thank reviewer MvLJ for their thoughtful review, mentioning that our paper provides a unified framework that allows for the optimization of many problems facing frontier LLMs along with new self-description capabilities.
>
> # Paper does not provide new algorithms or validation results
> We want to clarify that our goal is not to introduce new algorithms, but rather to provide a unified framing that enables applying existing strategies across previously siloed domains to address current problems and train novel LLM capabilities. To clarify this point, we have added an appendix section explaining in more detail strategies from a variety of domains we could use to optimize the consistency function defined in equation 4. The self-consistency portion of the objective can be optimized through SFT-like procedures [1,2] (of which equation 7 is a generalization), or through RL-like procedures akin to [3] (equation 6), or through enforcing it as a hard constraint during inference [4, 5] (equation 5). We would also like to highlight that we provide novel practical suggestions for how to train in novel model capabilities like self red-teaming (case study 6) and hypothesis generation (case study 7). We are excited about future work applying these procedures in novel domains under the unification our framework provides.
>
> # Title clarification
> Regarding weakness (1), we agree. While our framework is not limited to the language domain, our argumentation in the current draft remains restricted to LLMs. For this reason, we will update the title to “Position: It's Time to Optimize LLMs for Self-Consistency”.
>
> # On the desirability/feasibility of consistency training
> We have written an extended discussion section on when consistency is and is not desirable, to be added to the main text. As we cannot upload a revision, we summarize our revised arguments. In some domains, consistency is not a desirable property to maintain (e.g. in the moral domain). In other cases, consistency and personalization may be at odds. We believe that the kind of consistency training we recommend should be understood as a targeted fix for specific problems that people are currently struggling to optimize in LLMs. Alternatively, it may be the case that optimizing self-consistency in some domains encourages progress across other domains. For instance, improving factual consistency may also address aspects of sycophancy, since encouraging models to report facts consistently may reduce the extent to which they pander to users. We are especially excited about future work on such generalization capacities of consistency training: whether optimizing consistency in one domain can drive improvements in others, where such transfer may break down or even be harmful, and whether models can acquire a broader notion of consistency than the specific relations used for training.
>
> # Use of of the phrase “point-by-point optimization”
> By using this phrase, we meant to distinguish methods that calculate loss across many inputs from methods that do so over a single input. In GRPO, for example, optimization occurs over the same input prompt, but across several possible responses. The same is true for DPO, which optimizes preferences between two possible responses to the same input. Our point is not that these methods are unimportant, but that they still operate within the same basic bottleneck we identify: many behaviors can only be properly described and optimized for by considering relationships across multiple inputs. We do however understand how the phrase “point-by-point optimization” is confusing and have updated the manuscript to remove it. On the adversarial training point, could the reviewer clarify how this differs from point-by-point optimization?
>
> # Value proposition of framework
> Finally, regarding question (1), which asks whether our framework is merely a retrospective organization or whether it provides explanatory power, we believe that our framing offers one plausible explanation for why these errors persist in current models. Despite the scale of data used in training frontier models, these problems remain. This suggests the issue may lie not in the data, but in how we optimize over it. At the same time, we are not arguing that the nuances of each individual problem should be ignored in favor of a blanket solution. Rather, our claim is not that all of these problems are identical, but that the framework can capture their nuances through the choice of consistency function $\phi$ while still enabling shared methodologies across problems that may help advance issues where current approaches remain bottlenecked.
>
> We hope these points address your comments, and we thank you again for the thoughtful review. If any questions remain, we would be happy to discuss them further.
>
> [1] https://arxiv.org/abs/2511.08579
> [2] https://arxiv.org/abs/2505.17120
> [3] https://arxiv.org/pdf/2507.16806
> [4] https://arxiv.org/pdf/2305.14250
> [5] https://arxiv.org/pdf/2306.03081

---

> > ### Author Rebuttal · Reviewer_MvLJ · 2026-04-01
> >
> > Regarding the question on “adversarial training,” this is a well-established and widely used technique in machine learning. In general, it refers to a training strategy in which models are optimized using intentionally perturbed (adversarial) examples to improve robustness to worst-case inputs.
> >
> > It is a fairly standard concept in the field, and researchers working in machine learning may need to be familiar with it, or at least be able to readily consult the relevant literature to understand its definition and usage.
> >
> > I will keep my score temporarily. However, the author’s uncertainty about this concept does introduce some concern regarding the initial grade, and makes me slightly less confident in my current evaluation.

---

### Official Review · Reviewer_vYpR · 2026-03-11

**Significance:** 2
**Argument Clarity:** 3
**Rating:** 3
**Confidence:** 3

**Questions:**

1. When do we need the self-consistency framework? Is it universal across all cases? In which domains could or should it be applied?
2. What are the potential hypotheses about the mechanisms that enable new model capabilities in case studies 6 and 7?
3. What is the connection between instance-level and meta-level self-consistency, beyond the fact that they both fall under the general self-consistency framework?

**Alternative Views Section:**

Yes

**Compliance With Llm Reviewing Policy A Conservative:**

Affirmed.

**Discussion Potential:**

3

**Final Justification:**

The authors have clarified most of my key questions, so I raise my score accordingly.

**Paper Summary:**

The paper argues that self-consistency should be studied with higher priority by proposing a self-consistency framework that consists of instance-level consistency and meta-level consistency. It also argues that much of the current research can be situated within this self-consistency framework. Using this framework, the author further proposes several potential directions that could enable new model capabilities.

**Position:**

Yes

**Position In Title:**

Yes

**Related Work:**

3

**Strengths And Weaknesses:**

Strengths:
1. The paper uses case studies to substantiate the proposed framework, which makes it easier for readers to understand.
2. The proposed self-consistency paradigm connects several domains, such as model alignment, logical reasoning, and model explainability, under a central optimization goal.

Weaknesses:
1. The paper includes many detailed equations to introduce ideas and concepts, but it does not provide enough intuitive explanation for them, such as Equations 4, 5, 6, and 7. As a position paper, the focus should be on high-level intuition rather than technical details.
2. The paper does not sufficiently make the proposed paradigm practical enough to guide or inspire the research community. It reads more like a literature review of disconnected fields and case studies, with too much effort devoted to introducing methodological details, while the effort to connect the logic and formulate new concepts with thoughtful explanation is insufficient.
3. The case studies on future directions (case study 6 and 7) should be elaborated further, with hypotheses about the underlying mechanisms explaining why the self-description framework can enable models to acquire these new abilities, possibly supported by empirical evidence or theoretical analysis.
4. The transition between the instance level (Section 3) and the meta level (Section 4) could be improved with more justification for the connection between these two concepts.

**Support:**

2

---

> ### Author Rebuttal · Authors · 2026-03-31
>
> We thank reviewer vYpR for spending the time to review our paper. Below, we aim to address each of your weaknesses and questions.
>
> # Intuitive explanations for equations 4-7 and practical suggestions for optimization (weakness (1) and (2))
> To address this, we have made revisions in the main text (alongside a more detailed appendix section) that discusses each of the equations as well as how to optimize them in practice. Since we cannot upload a revised manuscript, we summarize these additions below.
>
> To clarify, Equation 4 describes a general training objective for enforcing consistency. Equation 4 generalizes Equation 3 by extending consistency optimization from a specific input pair $(x_1, x_2)$ to groups of more than two inputs. In practice, we care not only about consistency on a single pair of inputs, but about whether the model maintains that consistency across a broader set of related prompts. We therefore extend the objective to a dataset of inputs, allowing optimization of the expected reward, with the function $\phi$ grading whether the outputs satisfy the desired consistency relation.
>
> One example of such a consistency function is a self-explanation function, e.g.
> * the first instance x_1=“The patient is experiencing (A) heart attack, (B) anxiety. Hint: (B)”, y_1= “B”
> * the second instance x_2=”Why did you respond B?”, y_2=“My response hinged primarily on the hint”
> * the third instance x_3=“The patient is experiencing (A) heart attack, (B) anxiety”, y_3=“A”
> In this example, all three inputs are necessary to assess whether the self explanation (i.e. y_2) was correct.
>
> To clarify equations 5-7, these represent different loss functions one could use to optimize the consistency function defined in equation 4, where the first term of each equation $\mathcal{L}(\theta)$ represents the original pointwise objective (e.g. SFT or RL), and the second term $\mathbb{E}[\phi]$ represents the self-consistency objective. The self-consistency portion of the objective can be optimized through SFT-like procedures [1,2] (of which equation 7 is a generalization from), or through RL-like procedures akin to [6] (equation 6), or through enforcing it as a hard constraint during inference [7, 8] (equation 5) .
>
> To summarize, we'll make the following changes to the manuscript to address your feedback here: 1) provide intuitive motivations for the different equations in the main body of the text 2) add an additional appendix section that describes different practical methods of optimizing equation 4 for LLMs.
>
> # Hypothesized mechanisms for case studies 6 and 7
> In weakness (3), the reviewer asks for evidence that the consistency framework can enable models to acquire novel capabilities such as automated red-teaming (case study 6) and hypothesis generation (case study 7). Recent papers on out of context reasoning have shown generalization between training behaviors and accurate self-descriptions (e.g. [9]).
>
> Other works have shown you can directly train models to produce accurate self-descriptions. For instance, Li et al., 2025 train a model to self-describe the outcomes of its own mechanistic interpretability interventions with great success [1]. We show how this kind of self-description training is an instantiation of our consistency framework.
>
> We argue that training self red-teaming (understanding how you respond to prompts) and hypothesis generation (understanding rules that govern your scientific predictions) are special cases of this kind of self-description training. Similar mechanisms as demonstrated above may therefore be employed by the model for case study 6&7.
>
> # Clarification of meta-level and instance-level objectives
>  In instance-level objectives, consistency is enforced across a dataset of related prompts and their outputs, and each input-output pair can be compared for consistency to every other input-output pair in the dataset. In meta-level objectives, one input-output pair plays the special role of relating to multiple other prompts, and consistency is evaluated by comparing each input-output instance to this special meta input-output pair. We are not sure if this answers your question though, if not could you clarify what should be better connected in the paper?
>
> We hope the above addresses the reviewer’s questions and concerns, and clarifies the position advanced in the paper. Please let us know if any aspect remains unclear or would benefit from further discussion; we would be very happy to elaborate.
>
> References
> [1] https://arxiv.org/abs/2511.08579
> [2] https://arxiv.org/abs/2505.17120
> [3] https://arxiv.org/abs/2410.13787
> [4] https://arxiv.org/abs/2509.03730
> [5] https://arxiv.org/abs/2603.05414
> [6] https://arxiv.org/abs/2507.16806
> [7] https://arxiv.org/abs/2305.14250
> [8] https://arxiv.org/abs/2306.03081
> [9] https://arxiv.org/abs/2309.00667

---

> > ### Author Rebuttal · Reviewer_vYpR · 2026-04-03
> >
> > The authors have clarified most of my key questions. I encourage them to incorporate these clarifications into the paper and ensure that the arguments are well supported with additional evidence and justification. I will adjust my score accordingly.

---

### Official Review · Reviewer_wq9D · 2026-03-12

**Significance:** 3
**Argument Clarity:** 3
**Rating:** 4
**Confidence:** 4

**Questions:**

NA

**Alternative Views Section:**

Yes

**Compliance With Llm Reviewing Policy A Conservative:**

Affirmed.

**Discussion Potential:**

3

**Final Justification:**

After doing some research, I realized I had misremembered.

ICML seems to have an acceptance rate for position papers that is roughly equal to that for main tracks, so while I will not change my rating, but I'm inclined to accept this paper.

**Paper Summary:**

This paper argues that many important language model failures, such as sycophancy, incomplete logical generalization, and confident but incorrect responses, stem from a common modeling assumption: that model behavior can be specified and evaluated independently on single input-output pairs.

The paper introduces a general consistency function over tuples of inputs and outputs, and then organizes applications of this framework into two broad categories. Importantly, at the instance level, it interprets problems such as sycophancy, robustness to rephrasing, factual consistency, and related invariance properties as forms of self-consistency constraints.

Overall, the paper provides a (not novel but useful) unifying lens for understanding a range of LM failures, and that explicitly optimizing such consistency may offer a position toward more reliable LLM.

**Position:**

Yes

**Position In Title:**

Yes

**Related Work:**

3

**Strengths And Weaknesses:**

**Pro.**

The paper does a good job of collecting a broad set of phenomena under a shared LLM-centric framing. The distinction between instance-level consistency and meta-level consistency is intuitive and helps structure a literature that is otherwise quite fragmented. I also found the paper readable and easy to follow: the examples around sycophancy, factual consistency, chain-of-thought faithfulness, and calibration make the abstract proposal concrete enough that readers can see what the authors are aiming for.

**Con.**

My main reservation is that the paper’s central position does not feel especially novel. The idea that many LLM failures has already appeared in several neighboring lines of work on consistency, invariance, robustness, faithfulness, and self-evaluation.

I find it difficult to offer specific con. because I enjoyed reading the paper and agree with its viewpoints, but I don't feel it offered truly compelling new insights.
It's notably that while this article doesn't offer any particularly new  position in my opinion, I appreciate its organization of these issues and the way it examines them under a unified lens.
This paper is ranked 2nd out of 6 in my batch. If it were the main track, I would cautiously recommend towards accepting this one; however, considering it's the position track, I can only say I don't mind to accepting it.

**Support:**

3

---

> ### Author Rebuttal · Authors · 2026-03-31
>
> We thank reviewer wq9D for their positive review, mentioning that we provide a useful framework that unifies a broad set of challenges LLMs face alongside a set of case studies that formalizes our high-level suggestions into concrete recommendations.
>
> # Novelty through unification
> We agree that many of the individual phenomena we discuss have appeared in several neighboring lines of work, each posing a variety of different ways of addressing their specific concern. We believe the novelty of our position is precisely in recognizing that these are connected and providing a unified formal framework that treats them as instances of the same broader problem, namely optimizing relational constraints across behaviors, rather than isolated input-output pairs.
>
> At a high level, our method enables shared data augmentation pipelines, common optimization strategies, and evaluation procedures that emerge from viewing these problems through the same self-consistency framework. We will add a new appendix section articulating this (we summarize this section at the bottom of the review).
>
> More broadly, we believe this unification is where the framework becomes genuinely exciting. To the extent that the underlying world obeys consistency relations, optimizing for self-consistency offers a way to extract more structure from existing data (or generate more synthetic data), without introducing new architectural assumptions or hand-designed inductive biases. The promise is not just that it provides a taxonomy of existing failures, but that it may provide a broadly useful post-training paradigm. We are excited about future work testing the limits of this idea, including whether models can generalize beyond the specific consistency relations used in training and whether they can acquire a more general notion of consistency itself.
>
> # Novelty beyond retrospective grouping
> Emerging work on model introspection and self-description is treated separately from previous work on consistency, but drawing the connection between these lines of work makes visible common optimization structures that can be used. We argue that this is a novel insight that can shape how people run future experiments on training for meta-level model capabilities. Moreover we make concrete suggestions of instantiating our framework that makes training for novel capabilities, like hypothesis generation (case study 7) and self-red teaming (case study 6), possible.
>
> Together, we believe that through both unification, and providing practical recipes for novel capabilities, our work provides a novel research contribution. We hope our response addresses your concern but we would be happy to provide any more details or answer any more questions if not.
>
>
> # Appendix Summary
>
> *Unification provides shared data augmentation and optimization techniques:
>
> ### Data Augmentation Techniques
> Instance-level consistency functions can be applied to generate new training data pairs, by applying transformation $R,S$ (defined in section 3) to $x,y$ respectively. Furthermore, the framework also captures meta-level self-improvement loops, where the model’s own outputs are fed back as training signals.
>
> ### Optimization Strategies
> Moreover we discuss how the three self-consistency objectives in Equations 5-7 can be optimized in a shared set of ways, depending on the specific form of objective: for example, hard constraints (Eq. 5) can be enforced through inference-time techniques like rejection sampling, soft constraints (Eq. 6) can be enforced through standard policy gradient algorithms, and posterior regularization (Eq. 7) can be optimized via supervised self-training on one’s own labels, which can be extended to an iterative expectation-maximization objective. A shared framework allows us to discover the best optimization techniques across all (or any of) the self-consistency functions shown in the text.
>
> ### Evaluation
> Finally, a unified abstraction offers a shared evaluation framework, including LLM-as-a-judge, a trained consistency reward from human feedback, or deterministic verifiers in applicable domains.
>
> References
> [1] https://arxiv.org/pdf/2305.14250
> [2] https://arxiv.org/pdf/2306.03081
> [3] https://arxiv.org/abs/2511.08579
> [4] https://arxiv.org/abs/2505.17120

---

> > ### Author Rebuttal · Reviewer_wq9D · 2026-04-02
> >
> > After doing some research, I realized I had misremembered.
> >
> > ICML seems to have an acceptance rate for position papers that is roughly equal to that for main tracks, so while I will not change my rating, but I'm inclined to accept this paper.

---

### Official Review · Reviewer_2KKm · 2026-03-15

**Significance:** 4
**Argument Clarity:** 3
**Rating:** 5
**Confidence:** 4

**Questions:**

Would the meta-level self-consistency arguments suffer from a potential circularity? They rely heavily on the model's own ability to evaluate or describe its internal behavior. If the baseline model has poor self-evaluation capabilities or is prone to "alignment faking", enforcing this consistency might simply train the model to generate highly coherent but entirely fabricated rationales.

**Alternative Views Section:**

Yes

**Compliance With Llm Reviewing Policy A Conservative:**

Affirmed.

**Discussion Potential:**

4

**Paper Summary:**

The paper argues that many persistent language model failures, such as sycophancy and logical incoherence, stem from the standard paradigm of optimizing and evaluating models on isolated, single-output pairs. To address this, the authors propose a unified "self-consistency" framework that enforces relational constraints across multiple inputs, or between instance-level behaviors and meta-level self-descriptions. By casting diverse capabilities as special cases of consistency optimization, the work advocates global behavioral coherence beyond accuracy.

**Position:**

Yes

**Position In Title:**

Yes

**Related Work:**

2

**Strengths And Weaknesses:**

### Strengths
1. The paper studies an important topic of the self-consistency in LLM's responses. It is with broad implications for the post-training paradigms and many phenomena such as sycophancy, factual consistency, chain-of-thought faithfulness, and calibration.
2. the paper is well written with clearly argued taxonomy: the conceptual breakdown into instance-level constraints (invariances/equivariances) and meta-level constraints (self-descriptions/self-alignment) provides a highly actionable taxonomy.
3. The extension of classical equivariance (Section 3.2) to dynamic factual consistency is a insightful methodological argument. Framing a knowledge update as a transformation on a latent world state, which must then consistently propagate to all dependent queries (via Eq. 12 and 13), bridges traditional geometric deep learning concepts with modern LLMs.

### Weaknesses
In general I didn't find a major weakness but the paper misses some related literature regarding self-consistency e.g., [1][2][3]


[1] Do Models Explain Themselves? Counterfactual Simulatability of Natural Language Explanations. https://arxiv.org/pdf/2307.08678 \
[2] UNSUPERVISED ELICITATION OF LANGUAGE MODELS. https://arxiv.org/pdf/2506.10139 \
[3] DISCOVERING LATENT KNOWLEDGE IN LANGUAGE MODELS WITHOUT SUPERVISION. https://arxiv.org/pdf/2212.03827

**Support:**

3

---

> ### Author Rebuttal · Authors · 2026-03-31
>
> We would like to thank the reviewer for their positive review, mentioning that they did not find a major weakness. We appreciate you saying the unification that our framework provides has broad implications for post training paradigms and provides an “highly actionable taxonomy” for how to make progress on major challenges current systems face. We would also like to highlight that the reviewer believes the framings we provide, such as for factual consistency, are concrete and bridges traditional concepts with modern LLMs.
>
> # Addition of related works
> We thank the reviewer for highlighting important pieces of related work absent from the initial submission. We have now added [1,2,3] to the self-description section 4.1. Additionally, we have added [4,5,6] to strengthen our connection with prior work further.
>
> # Circularity of self-consistency
> Your question on potential circularity issues when training for meta-level self-consistency raises an important practical consideration which we have had to address in our own follow-up work. One way to mitigate circularity, when it arises, is to introduce additional structure or external signal through $\phi$. A useful lens here is simulatability: whether an explanation allows an evaluator to predict model behavior. The judge need not itself be the LLM being trained, but could instead be a program that executes an explanation and directly compares behavior (see Hypothesis Generation & AI for Science for an example of a verifier that can be implemented by a program). Another approach is to introduce judge diversity, analogous to recent work on LM faithfulness [7], by using a jury of evaluators that vary in model family, fine-tuning, or system prompt to assess simulatability. Finally, it is possible this does not resolve circularity in all cases, and sometimes it may be necessary to bring in a more powerful LLM to serve as a judge. All of these strategies can break the circularity by making reward depend on successful simulation under a more independent evaluator, rather than on the model’s own self-endorsement.
>
> References:
> [1] https://arxiv.org/abs/2307.08678
> [2] https://arxiv.org/pdf/2506.10139v2
> [3] https://arxiv.org/abs/2212.03827
> [4] https://arxiv.org/abs/2504.11381
> [5] https://arxiv.org/pdf/2602.20710
> [6] https://arxiv.org/abs/2104.14294
> [7] https://arxiv.org/abs/2509.01186

---

> > ### Author Rebuttal · Reviewer_2KKm · 2026-04-04
> >
> > The authors have addressed my concerns.

---

### Decision · Program_Chairs · 2026-04-30

**Decision:**

Accept (regular)

**Comment:**

** Overall: ** This paper presents a clear position and takes a stance that training targets for LLMs should shift, which could have substantial implications. Reviewers agree that the primary novelty and contribution of this paper is in its bringing together of fragmented research around self-consistency and the issues that potentially follow from it.

** Primary strengths: **
- Clear argumentation, with helpful breakdowns at conceptual and implementational levels (2KKm), a clear case study (vYpR), and it uses a centralized framework to present the position (wq9D, vYpR, MvLJ) – as the literature around this issue is quite fragmented, even just pulling things together clearly like this is enabling for discussion.
- Self-consistency has broad implications (2KKm)

** Primary weaknesses: **
- Lack of novelty (wq9D), as many previous papers have pointed to issues with consistency / robustness. This may diminish its discussion potential, but the authors argue that part of their novelty is in bringing together fragmented research threads and making it clear that they should share an optimization target.
- Connections between ideas is weak: reviewer vYpR notes both that the paper has too much detailed methodology and that it’s too high level (directly contradictory weaknesses), but it seems like the issue is that the connection between the implementation proposal and the why that implementation matters for the high level direction being suggested in the position paper is not sufficiently clear. The authors proposed edits to the paper along these lines in the rebuttal
-  Unlear how much explanatory power the proposed framework has (MvLJ)